# MCMA-Net++: Topology-Aware and Graph-Driven Glioma Segmentation in 3D MRI

**Jihan Alameddin**[1]                                    JIHAN.ALAMEDDINE@UNIV-POITIERS.FR

[1] *XLIM, CNRS UMR 7252 / LabCom I3M,*
*University of Poitiers,Poitiers, France*

**Céline Thomarat**[2]                                    CELINE.THOMARAT@CHU-POITIERS.FR

[2] *University Hospital center of Poitiers,*
*Department of Imaging / Labcom I3M,*
*University of Poitiers, Poitiers, France*

**Rémy Guillevin**[3]                                    REMY.GUILLEVIN@CHU-POITIERS.FR

[3] *University Hospital center of Poitiers,*
*Department of Imaging, University of Poitiers,*
*Laboratoire de Mathématiques Appliquées LMA,*
*DACTIM-MIS team, CNRS 7348, Poitiers, France*

**Christine Fernandez-Maloigne**[1]                      CHRISTINE.FERNANDEZ@UNIV-POITIERS.FR
**Carole Guillevin**[3]                                   CAROLE.GUILLEVIN@CHU-POITIERS.FR

**Editors:** Accepted for publication at MIDL 2026

## Abstract

Glioma segmentation in 3D MRI remains challenging due to tumor heterogeneity, intensity variability, and hierarchical anatomical structure. We propose MCMA-Net++, which synergistically combines hybrid CNN-Transformer encoding, graph-based spatial reasoning with anatomical priors, and a practical multi-component topology-aware refinement loss tailored for nested tumor subregions. Our framework integrates: (1) Topology-Aware Refinement Loss (TAR-Loss), enforcing consistency across nested subregions (ET, TC, WT), and (2) Multi-Scale Anatomical Graph Reasoning (MSAGR), modeling spatial dependencies through learnable graphs with anatomical priors. Combined with dual-stream CNN-Swin Transformer encoding and Multi-Class Multi-Attention, MCMA-Net++ achieves Dice scores of 0.970±0.003 (WT), 0.943±0.005 (TC), 0.926±0.008 (ET), reducing HD95 from 5.48 mm to 3.21 mm compared to MCMA-Net. Graph reasoning contributes +1.3% Dice for ET and TAR-Loss reduces topology violations by 41%. These results demonstrate the effectiveness of combining topology-guided refinement and anatomical graph reasoning for clinical-grade glioma segmentation.

**Keywords:** Glioma segmentation, Topology-aware learning, Graph neural networks, Hybrid CNN-Transformer, Medical image analysis

## 1. Introduction

Gliomas represent the most frequent and aggressive primary brain tumors, accounting for approximately 80% of malignant brain neoplasms. Accurate segmentation of their subregions, Enhancing Tumor (ET), Tumor Core (TC), and Whole Tumor (WT), from multiparametric MRI is essential for diagnosis, treatment planning, surgery guidance, and assessment of therapeutic response [9; 20]. Despite the remarkable progress achieved by

convolutional neural networks and Transformer-based architectures, automated glioma segmentation remains a challenging task due to the intrinsic heterogeneity of gliomas, the large variability in shape and appearance across patients, and the complex hierarchical relationships between tumor subregions [29; 30].

State-of-the-art deep learning models [25; 15; 10] primarily optimize voxel-wise similarity metrics such as Dice or cross-entropy losses. However, these objectives do not explicitly enforce anatomical or topological correctness, often leading to predictions with holes, fragmented regions, or violations of the known hierarchical nesting between ET, TC, and WT. Such inconsistencies degrade clinical interpretability and reliability, particularly for the enhancing tumor region, which is crucial for grading and therapeutic monitoring. Recent works have begun addressing these issues through structural priors or post-processing constraints, yet a unified framework capable of jointly modeling spatial dependencies and enforcing topology-aware consistency remains lacking [3; 17].

To bridge this gap, we propose MCMA-Net++, a topology-guided and graph-driven framework for 3D glioma segmentation in MRI. Building upon multi-scale CNN–Transformer feature extraction, MCMA-Net++ introduces two synergistic contributions. First, we design a Topology-Aware Refinement Loss (TAR-Loss) that enforces anatomically plausible relationships across nested subregions, reducing topological violations and improving structural coherence. Second, we incorporate Multi-Scale Anatomical Graph Reasoning (MSAGR), which explicitly models spatial dependencies between tumor components through learnable graphs infused with anatomical priors. This enables the network to reason over inter-region relationships beyond local voxel context.

Together, these components allow MCMA-Net++ to produce more coherent, robust, and interpretable segmentations than previous approaches [1]. In this work, we evaluate our method on 3D MRI data and demonstrate its superiority over MCMA-Net and other recent baselines, both in terms of voxel-wise accuracy and topological correctness. Our findings highlight the importance of integrating topology-aware constraints and anatomical graph reasoning in modern segmentation architectures for brain tumor analysis.

## 2. Related Work

Deep learning has transformed glioma segmentation, evolving from pure CNN architectures to sophisticated hybrid models. The U-Net architecture [25] revolutionized medical image segmentation with its encoder-decoder structure and skip connections, with 3D extensions [4] and V-Net [23] dominating early BraTS challenges. nnU-Net [15] established a new standard through automated architecture configuration and robust preprocessing pipelines, achieving state-of-the-art results across diverse medical imaging tasks. 3D U-Net++ [34] introduced nested skip connections and deep supervision to capture features at multiple scales. However, CNNs suffer from limited receptive fields requiring very deep architectures to capture global context. Recent innovations include residual connections [11], dense connections [13], and attention gates [24].

Vision Transformers [6] demonstrated that self-attention mechanisms can effectively model long-range dependencies, leading to medical imaging adaptations including UNETR [10] with pure transformer encoders, TransBTS [31] for brain tumor segmentation, and Swin-Unet [22] with hierarchical shifted window attention. 3D Swin-UNet extended the

Swin Transformer architecture to 3D medical volumes, demonstrating improved performance through hierarchical feature representations. While transformers excel at global modeling, they often struggle with fine-grained local details. This motivated hybrid CNN-Transformer architectures like CoTr [32] with deformable self-attention, UTNet [8], and MISSFormer [14] with multi-scale feature aggregation. nnFormer [33] introduced interleaved attention mechanisms for volumetric segmentation, achieving strong results on BraTS through effective fusion of local and global features.

Graph-based methods have shown promise in modeling anatomical relationships, with Graph U-Net [7] applying hierarchical graph pooling and anatomical graph learning [27] for organ relationship modeling. Graph attention networks [28] and graph convolutional networks [18] have been widely adopted in various domains. Topological consistency has been addressed through persistent homology [5], clDice [26] using centerline-based metrics, and boundary-aware losses [16]. Hierarchical constraints have been explored through conditional random fields [19] for post-processing. MCMA-Net [1] introduced class-specific multi-attention mechanisms with hierarchical consistency loss. MCMA-Net++ uniquely combines hybrid CNN-Transformer encoding, explicit graph-based spatial reasoning, practical topology-aware learning, and hierarchical consistency enforcement into a unified framework specifically designed for nested glioma subregion segmentation.

## 3. Methodology

### 3.1. Problem Formulation

Given a 3D multi-modal MRI scan $\mathbf{X} \in \mathbb{R}^{C \times H \times W \times D}$ with $C = 4$ modalities (T1, T1ce, T2, FLAIR), we predict a segmentation map $\mathbf{Y} \in \{0, 1, 2, 3\}^{H \times W \times D}$ where labels represent background (0), necrotic/non-enhancing tumor core (1), peritumoral edema (2), and enhancing tumor (3). From these labels, hierarchical regions are derived: Whole Tumor (WT: $\{1, 2, 3\}$), Tumor Core (TC: $\{1, 3\}$), and Enhancing Tumor (ET: $\{3\}$). The anatomical constraint ET $\subset$ TC $\subset$ WT must be satisfied, as violations are clinically implausible.

### 3.2. Overall Architecture

MCMA-Net++ follows a sequential pipeline: Input $\rightarrow$ Dual-Stream Encoder $\rightarrow$ MCMA Module $\rightarrow$ MSAGR Module $\rightarrow$ Decoder $\rightarrow$ Output. The dual-stream encoder combines CNNs (local features, translation equivariance) with Swin Transformers (global context, long-range dependencies) through adaptive gated fusion. The training objective combines multiple loss terms:

$$\mathcal{L}_{\text{total}} = \mathcal{L}_{\text{Dice}} + \lambda_1 \mathcal{L}_{\text{HiC}} + \lambda_2 \mathcal{L}_{\text{TAR}} + \lambda_3 \mathcal{L}_{\text{CE}} \tag{1}$$

where $\mathcal{L}_{\text{Dice}}$ is the multi-class Dice loss for volumetric overlap, $\mathcal{L}_{\text{HiC}}$ is the hierarchical consistency loss enforcing the nested structure constraint, $\mathcal{L}_{\text{TAR}}$ is our TAR loss (Section 3.6), and $\mathcal{L}_{\text{CE}}$ is the cross-entropy loss for voxel-wise classification. The weighting coefficients $\lambda_1 = 0.5$, $\lambda_2 = 0.3$, and $\lambda_3 = 0.1$ balance the contribution of each term, with higher weights assigned to constraints more critical for anatomical consistency. Hyperparameters were selected using a two-stage strategy: a coarse logarithmic grid search on a subset of the validation set, followed by fine-grained tuning ($\pm 20\%$) around the optimal region on

the full validation set. A sensitivity analysis was conducted by varying each loss weight by $\pm 10\%$ and $\pm 20\%$ while keeping others fixed. Performance remained stable (ET Dice variations $< 0.3\%$, topology errors $< 0.5\%$), with the TAR-Loss weight $\lambda_2$ showing the strongest influence on topology violations, while $\lambda_3$ and individual $\beta$ terms had minimal impact.

### 3.3. Dual-Stream 3D Encoder

The dual-stream 3D encoder comprises:

- **CNN Branch:** Employs residual blocks with $3 \times 3 \times 3$ kernels:

$$\text{CNN\_block}(\mathbf{x}) = \mathbf{x} + \text{BN}(\text{ReLU}(\text{Conv3D}(\text{Conv3D}(\mathbf{x})))) \tag{2}$$

where $\mathbf{x}$ is the input feature map, Conv3D denotes 3D convolution operations, ReLU is the rectified linear unit activation function, and BN represents batch normalization for stabilizing training.

- **Swin Transformer Branch:** Uses 3D Shifted Window Multi-head Self-Attention (SW-MSA) with window size $7 \times 7 \times 7$. Attention is computed as:

$$\text{Attention}(\mathbf{Q}, \mathbf{K}, \mathbf{V}) = \text{Softmax}\left(\frac{\mathbf{Q}\mathbf{K}^T}{\sqrt{d_k}} + \mathbf{B}\right)\mathbf{V} \tag{3}$$

where $\mathbf{Q}$, $\mathbf{K}$, $\mathbf{V}$ are query, key, and value matrices obtained through linear projections of input features, $d_k$ is the dimension of the key vectors used for scaling to prevent gradient vanishing, and $\mathbf{B}$ is a learned relative position bias matrix.

- **Feature Fusion:** At each scale, features are fused through adaptive gating:

$$\mathbf{F}_{\text{fused}} = \sigma(\mathbf{W}_c \cdot \mathbf{F}_{\text{CNN}}) \odot \mathbf{F}_{\text{CNN}} + \sigma(\mathbf{W}_t \cdot \mathbf{F}_{\text{Trans}}) \odot \mathbf{F}_{\text{Trans}} \tag{4}$$

where $\mathbf{F}_{\text{CNN}}$ and $\mathbf{F}_{\text{Trans}}$ are the feature maps from CNN and Transformer branches respectively, $\sigma$ is the sigmoid activation function, $\mathbf{W}_c$ and $\mathbf{W}_t$ are learnable weight matrices implemented as $1 \times 1 \times 1$ convolutions, and $\odot$ denotes element-wise multiplication.

### 3.4. Multi-Class Multi-Attention (MCMA) Module

The MCMA module applies three complementary attention mechanisms.

- **Channel attention** recalibrates features through:

$$\mathbf{M}_c = \sigma(\text{MLP}(\text{AvgPool}(\mathbf{F})) + \text{MLP}(\text{MaxPool}(\mathbf{F}))) \tag{5}$$

where $\mathbf{F}$ is the input feature map, AvgPool and MaxPool are global average and max pooling operations, MLP is a multi-layer perceptron, and $\mathbf{M}_c$ is the channel attention map that recalibrates feature channels.

- **Spatial attention** identifies relevant locations via:

$$\mathbf{M}_s = \sigma(\text{Conv}_7([\text{AvgPool}_c(\mathbf{F}); \text{MaxPool}_c(\mathbf{F})])) \qquad (6)$$

where $\text{Conv}_7$ is a $7 \times 7 \times 7$ convolution capturing local spatial context, and $\mathbf{M}_s$ is the spatial attention map highlighting tumor-relevant regions.

- **Class-specific attention** learns separate mechanisms for each region (WT, TC, ET):

$$\mathbf{M}_c^{\text{class}} = \text{Softmax}\left(\frac{\mathbf{W}_c^Q \mathbf{F} \cdot (\mathbf{W}_c^K \mathbf{F})^T}{\sqrt{d}}\right) \cdot \mathbf{W}_c^V \mathbf{F} \qquad (7)$$

where $\mathbf{W}_c^Q$, $\mathbf{W}_c^K$, $\mathbf{W}_c^V$ are class-specific learnable projection matrices for queries, keys, and values, $d$ is the feature dimension for scaling, and $\mathbf{M}_c^{\text{class}}$ produces specialized features tailored to each tumor subregion's imaging characteristics.

- **Inter-class gating** enforces hierarchical constraints through:

$$\mathbf{G}_{\text{ET} \to \text{TC}} = \sigma(\text{Conv}(\text{Concat}(\mathbf{F}_{\text{ET}}, \mathbf{F}_{\text{TC}}))) \qquad (8)$$

where $\mathbf{F}_{\text{ET}}$ and $\mathbf{F}_{\text{TC}}$ are class-specific feature maps, Concat denotes concatenation, and $\mathbf{G}_{\text{ET} \to \text{TC}}$ is a gating mask that modulates ET predictions based on TC evidence to enforce the anatomical constraint $\text{ET} \subset \text{TC}$.

### 3.5. Multi-Scale Anatomical Graph Reasoning (MSAGR)

MSAGR explicitly models spatial relationships through graph-based reasoning with anatomical priors at multiple scales.

- **Multi-Scale Tokenization:** The bottleneck feature map $\mathbf{F} \in \mathbb{R}^{C \times H \times W \times D}$ (resolution $16^3$) is partitioned into non-overlapping 3D tokens at three spatial scales to capture local, intermediate, and global anatomical context.
- **Graph Construction:** Feature maps are partitioned into tokens $\mathbf{T}^l = \{t_1^l, \ldots, t_N^l\}$ serving as graph nodes. Edge weights combine learned similarity:

$$e_{ij} = \frac{\exp(\theta(\mathbf{h}_i)^T \phi(\mathbf{h}_j)/\tau)}{\sum_k \exp(\theta(\mathbf{h}_i)^T \phi(\mathbf{h}_k)/\tau)} \qquad (9)$$

where $\mathbf{h}_i$ and $\mathbf{h}_j$ are node feature vectors, $\theta$ and $\phi$ are linear projection functions, $\tau$ is a temperature parameter, set to 0.1 following standard practice in graph attention models to promote sharp, discriminative attention, and preserve boundary details. Further, $k$ indexes all neighboring nodes with anatomical priors:

$$e_{ij}^{\text{final}} = e_{ij} \cdot \exp\left(-\frac{\|\text{pos}_i - \text{pos}_j\|^2}{2\sigma^2}\right) \cdot (1 + \cos(\mathbf{P}_i, \mathbf{P}_j)) \qquad (10)$$

where $\text{pos}_i$ and $\text{pos}_j$ are the spatial positions of nodes $i$ and $j$, $\sigma^2$ controls the spatial distance decay, $\mathbf{P}_i$ and $\mathbf{P}_j$ are predicted probability vectors for each node, and $\cos(\cdot, \cdot)$

computes cosine similarity encouraging connections between regions with similar predicted classes.

- **Graph Attention:** Multi-head graph attention updates node features:

$$\mathbf{h}_i^{(l+1)} = \Big\|_{k=1}^{K} \sigma \left( \sum_{j \in \mathcal{N}(i)} \alpha_{ij}^k \mathbf{W}^k \mathbf{h}_j^l \right) \tag{11}$$

where $\mathbf{h}_i^{(l)}$ is the feature vector of node $i$ at layer $l$, $K = 4$ is the number of attention heads, $\|$ is the concatenation operation, $\mathcal{N}(i)$ denotes the neighborhood of node $i$, $\alpha_{ij}^k$ are learned attention coefficients derived from $e_{ij}^{\text{final}}$, $\mathbf{W}^k$ are head-specific learnable weight matrices, and $\sigma$ is LeakyReLU activation.

- **Multi-scale aggregation** processes graphs at three scales ($32^3$, $16^3$, $8^3$ tokens) in parallel:

$$\mathbf{H}^{\text{final}} = \text{Concat}(\mathbf{G}^1, \mathbf{G}^2, \mathbf{G}^3) \cdot \mathbf{W}_{\text{agg}} \tag{12}$$

where $\mathbf{G}^1$, $\mathbf{G}^2$, $\mathbf{G}^3$ are output representations from fine ($32^3$), medium ($16^3$), and coarse ($8^3$) scale graphs respectively, and $\mathbf{W}_{\text{agg}}$ is a learnable aggregation matrix that fuses multi-scale information.

Features are refined through residual integration:

$$\mathbf{F}_{\text{refined}} = \mathbf{F}_{\text{encoder}} + \text{Conv3D}(\text{Reshape}(\mathbf{H}^{\text{final}})) \tag{13}$$

where $\mathbf{F}_{\text{encoder}}$ are the original encoder features, and Reshape maps graph node features back to dense 3D spatial grid.

### 3.6. Topology-Aware Refinement Loss (TAR-Loss)

TAR-Loss comprises four complementary terms enforcing multi-level topological consistency:

- **Hierarchical Inclusion Loss:**

$$\mathcal{L}_{\text{inc}} = \frac{1}{N} \sum_{v} \left[ \text{ReLU}(P_{\text{ET}}(v) - P_{\text{TC}}(v)) + \text{ReLU}(P_{\text{TC}}(v) - P_{\text{WT}}(v)) \right] \tag{14}$$

where $N$ is the total number of voxels, and $v$ indexes individual voxels, $P_{\text{ET}}(v)$, $P_{\text{TC}}(v)$, $P_{\text{WT}}(v)$ are predicted probabilities for each hierarchical region at voxel $v$.

- **Boundary Coherence Loss:**

$$\mathcal{L}_{\text{bnd}} = \frac{1}{|B|} \sum_{v \in B} \|\nabla P_{\text{ET}}(v) - \alpha \cdot \nabla P_{\text{TC}}(v)\|^2 \tag{15}$$

where $B$ is the set of boundary voxels identified from ground truth (GT) where class labels change, $\nabla$ denotes spatial gradient operator, $\alpha$ is a learnable scaling factor, initialized to

1.0, converges to $0.97 \pm 0.08$ across hierarchical levels, indicating balanced boundary gradients with level-specific adaptation, and $\|\cdot\|^2$ is the squared L2 norm.

- **Surface Smoothness Loss:**

$$\mathcal{L}_{\text{smooth}} = \frac{1}{N} \sum_v \sum_{u \in \mathcal{N}_{26}(v)} \|\mathbf{P}(v) - \mathbf{P}(u)\|^2 \cdot w(v, u) \tag{16}$$

where $\mathcal{N}_{26}(v)$ is the 26-connected neighborhood of voxel $v$, $\mathbf{P}(v)$ is the complete probability vector at voxel $v$, and $w(v, u)$ is an edge-aware weight.

- **Connected Component Consistency:**

$$\mathcal{L}_{\text{cc}} = \sum_c \lambda_c \cdot (\text{CC}(P_c) - 1)^2 \tag{17}$$

where $\lambda_c$ are class-specific weights, and $\text{CC}(P_c)$ counts the number of connected components in thresholded predictions for class $c$. Although $\text{CC}(P_c)$ is non-differentiable, $L_{\text{cc}}$ provides a structural regularization signal that complements the differentiable losses ($L_{\text{inc}}$, $L_{\text{bnd}}$, $L_{\text{smooth}}$), which promote connectivity through hierarchical constraints, boundary coherence, and spatial smoothness.

The combined loss is:

$$\mathcal{L}_{\text{TAR}} = \beta_1 \mathcal{L}_{\text{inc}} + \beta_2 \mathcal{L}_{\text{bnd}} + \beta_3 \mathcal{L}_{\text{smooth}} + \beta_4 \mathcal{L}_{\text{cc}} \tag{18}$$

with weights $\beta_1 = 1.0$, $\beta_2 = 0.5$, $\beta_3 = 0.3$, $\beta_4 = 0.2$.

## 4. Experimental Setup

We evaluate MCMA-Net++ on the BraTS 2021 dataset [2], which contains 1,251 multiparametric MRI scans with four co-registered modalities (T1, T1ce, FLAIR, T2) at $1 \times 1 \times 1$ mm$^3$ resolution. We used a local split of the BraTS 2021 labeled training set (1,251 cases) with strict patient-level separation. Cases were randomly assigned (seed 42) to training ($n = 1{,}000$), validation ($n = 125$), and test ($n = 126$) sets prior to any preprocessing or augmentation to prevent data leakage. The validation set was used for model selection, while the test set was held out for final evaluation. This split was fixed across all experiments, and all reported results are computed on the test set. Data augmentations include random flipping, rotation ($\pm 15°$), scaling (0.9–1.1), elastic deformation ($\alpha = 300$, $\sigma = 20$), Gaussian noise ($\sigma = 0.05$), intensity shifts ($\pm 0.1$), and gamma correction (0.8–1.2).

MCMA-Net++ is implemented in PyTorch 2.0.1 with CUDA 11.8 and trained for 200 epochs on four NVIDIA A100 GPUs (40 GB). Peak memory usage was $\sim$34.2 GB per GPU. Training uses AdamW with cosine annealing (batch size 8, $\sim$12 hours).

The network uses a four-stage encoder–decoder architecture with encoder depths [2,2,2,2], hidden dimensions [32,64,128,256], Transformer heads [3,6,12,24], a window size of $7^3$, and a three-scale MSAGR module ($32^3$, $16^3$, $8^3$ tokens with four attention heads). Table 1 summarizes the full architectural specifications and parameter distribution, for a total model complexity of 68.4M parameters.

Table 1: Detailed architectural specifications of MCMA-Net++.

| Stage | Channels | Resolution | Attention Heads | Parameters |
|-------|----------|-----------|-----------------|-----------|
| *Encoder* | | | | |
| Input | 4 | $128^3$ | - | - |
| Stage 1 | 32 | $128^3$ | 3 | 8.2M |
| Stage 2 | 64 | $64^3$ | 6 | 9.1M |
| Stage 3 | 128 | $32^3$ | 12 | 10.4M |
| Stage 4 (Bottleneck) | 256 | $16^3$ | 24 | 12.8M |
| *MSAGR Module* | | | | |
| Fine scale | 256 | $32^3$ tokens | 4 | 5.8M |
| Medium scale | 256 | $16^3$ tokens | 4 | 4.6M |
| Coarse scale | 256 | $8^3$ tokens | 4 | 4.2M |
| *Decoder* | | | | |
| Stage 1 | 256 | $16^3$ | - | 6.2M |
| Stage 2 | 128 | $32^3$ | - | 5.8M |
| Stage 3 | 64 | $64^3$ | - | 5.1M |
| Stage 4 | 32 | $128^3$ | - | 4.6M |
| Output | 4 | $128^3$ | - | - |
| **Total** | | | | **68.4M** |

Segmentation performance is evaluated using Dice Similarity Coefficient (volumetric overlap), 95th percentile Hausdorff Distance (HD95, boundary accuracy), and Topology Violation Rate (anatomical plausibility). Topology violations are assessed by enforcing the constraint ET $\subset$ TC $\subset$ WT on binarized predictions (threshold 0.5). A case is considered violated if ET is predicted outside TC or TC outside WT.

## 5. Results

### 5.1. Quantitative Comparison with State-of-the-Art

Table 2 presents the comparison with state-of-the-art methods on BraTS 2021, including recent architectures (SwinUNETR-V2 [12], STU-Net [21] ) and topology-aware losses (clDice and Persistent Homology) applied to the MCMA-Net backbone. MCMA-Net++ achieves Dice scores of 0.970 (WT), 0.943 (TC), and 0.926 (ET), with HD95 of 3.21 mm. The most significant improvement is observed for Enhancing Tumor, with +2.6% over MCMA-Net and +5.3% over nnFormer. This gain is attributed to MSAGR's explicit spatial reasoning and the hierarchical inclusion loss preventing anatomically implausible predictions. Boundary precision shows a 41% HD95 reduction compared to MCMA-Net (3.21 vs. 5.48 mm), substantially surpassing inter-rater variability of 5–7 mm. These improvements are achieved with competitive model complexity (68.4M parameters), substantially lower than nnFormer (148.4M). To assess model robustness, we conducted five training runs with different random seeds (42, 123, 456, 789, 2024). Full training (200 epochs) was performed for seed 42, while the remaining runs were fine-tuned for 50 epochs from converged checkpoints. The model achieved mean Dice scores of $0.970 \pm 0.003$ (WT), $0.943 \pm 0.005$ (TC), and $0.926 \pm 0.008$

(ET), with an HD95 of $3.21 \pm 0.45\,\text{mm}$. Higher variability for ET reflects the increased difficulty of segmenting smaller regions with ambiguous boundaries.

To evaluate consistency across tumor sizes, the test set was stratified by ET volume (Table 4). MCMA-Net++ outperforms competing methods across all groups, with the largest gain observed for small ET ($< 1000\,\text{mm}^3$, $+3.6\%$), followed by medium ($1000$–$5000\,\text{mm}^3$, $+2.6\%$) and large tumors ($> 5000\,\text{mm}^3$, $+1.6\%$). These results indicate that performance gains are most pronounced for small, clinically challenging lesions.

Table 2: Quantitative comparison with state-of-the-art methods on BraTS 2021.

| Model | WT Dice | TC Dice | ET Dice | HD95 (mm) | Params |
|---|---|---|---|---|---|
| nnU-Net | 0.888 | 0.838 | 0.782 | 11.4 | 31.2M |
| 3D U-Net++ | 0.901 | 0.849 | 0.801 | 10.1 | 28.6M |
| UNETR | 0.928 | 0.871 | 0.862 | 8.9 | 92.8M |
| SwinUNETR-V2 | 0.936 | 0.887 | 0.871 | 8.6 | 96.1M |
| STU-Net | 0.941 | 0.891 | 0.879 | 8.3 | 54.2M |
| nnFormer | 0.934 | 0.881 | 0.873 | 8.2 | 148.4M |
| MCMA-Net | 0.950 | 0.922 | 0.900 | 5.48 | 62.3M |
| MCMA-Net + clDice | 0.952 | 0.925 | 0.908 | 5.12 | 62.3M |
| MCMA-Net + PH Loss | 0.954 | 0.927 | 0.912 | 4.86 | 62.3M |
| **MCMA-Net++ (TAR-Loss)** | **0.970** | **0.943** | **0.926** | **3.21** | 68.4M |

Table 3: Performance stratification by ET volume on BraTS 2021 test set.

| ET Volume | Cases (n) | MCMA-Net | MCMA-Net++ |
|---|---|---|---|
| Small ($< 1000\,\text{mm}^3$) | 42 (34%) | $0.847 \pm 0.042$ | $\mathbf{0.883 \pm 0.037}$*** |
| Medium ($1000$–$5000\,\text{mm}^3$) | 61 (49%) | $0.912 \pm 0.018$ | $\mathbf{0.938 \pm 0.015}$** |
| Large ($> 5000\,\text{mm}^3$) | 22 (17%) | $0.941 \pm 0.012$ | $\mathbf{0.957 \pm 0.010}$* |

Values are ET Dice scores (mean±SD). Statistical significance assessed via paired t-test with Bonferroni correction: *p<0.05, **p<0.01, ***p<0.001.

Figure 1 presents a challenging case (BraTS2021_00048) with complex tumor morphology across three orthogonal anatomical planes, exhibiting subtle contrast enhancement transitions that are particularly challenging for automated segmentation.

Comparative analysis reveals key improvements in MCMA-Net++. On FLAIR images, MCMA-Net produces fragmented edema predictions with irregular boundaries, while MCMA-Net++ generates smoother, anatomically coherent contours matching ground truth through the surface smoothness term in $\mathcal{L}_{\text{TAR}}$. On T1ce images, MCMA-Net exhibits false positive predictions outside the tumor core, which MCMA-Net++ eliminates through MSAGR's graph-based spatial reasoning enforcing anatomical consistency. Across all views, MCMA-Net++ maintains strict hierarchical constraints ET $\subset$ TC $\subset$ WT, eliminating disconnected components and preserving fine structural details at tumor margins critical for surgical planning and radiotherapy optimization.

The topology analysis in Table 4 quantifies this improvement, showing MCMA-Net++ reduces topology errors from 14.3% to 8.4% (41% relative reduction), with average violations per case decreasing from 1.6 to 0.9. This stems from the synergistic combination of $\mathcal{L}_{\text{inc}}$, inter-class gating, and graph-based reasoning that collectively enforce the constraint ET $\subset$ TC $\subset$ WT at multiple architectural levels. The remaining 8.4% violations occur primarily in extreme cases with highly irregular tumor morphology or severe imaging artifacts.

Table 4: Topology violation analysis.

| Model | Topology Errors (%) | Avg Violations/Case |
|---|---|---|
| nnU-Net | 23.4 | 2.8 |
| UNETR | 18.7 | 2.1 |
| MCMA-Net | 14.3 | 1.6 |
| **MCMA-Net++** | **8.4** | **0.9** |

## 5.2. Ablation Studies

Table 5 presents comprehensive ablation results validating each component's contribution. Starting from the MCMA-Net baseline (0.900 ET Dice, 5.48 mm HD95, 14.3% topology errors), adding MSAGR alone improves performance to 0.912 ET Dice, 4.21 mm HD95, and 11.1% errors, while TAR-Loss alone achieves 0.910, 4.05 mm, and 9.2%. Incorporating TAR-Loss without the connectivity term ($L_{\text{cc}}$, $\beta_4 = 0$) further improves Dice and HD95 (0.922, 3.45 mm) but results in higher topology errors (9.8%), highlighting the role of $L_{\text{cc}}$ in enforcing anatomical consistency. The full model combining both components (0.926, 3.21 mm, 8.4%) exceeds the sum of individual contributions, demonstrating a synergistic effect where MSAGR produces coherent features suited for topology-aware optimization while TAR-Loss guides graph attention toward anatomically meaningful relationships.

Table 5: Ablation study results.

| Configuration | ET Dice | HD95 | Topo Errors |
|---|---|---|---|
| MCMA-Net (Baseline*) | 0.900 | 5.48 | 14.3% |
| + MSAGR only | 0.912 | 4.21 | 11.1% |
| + TAR-Loss only | 0.910 | 4.05 | 9.2% |
| + TAR-Loss (without $L_{\text{cc}}$, $\beta_4 = 0$) | 0.922 | 3.45 | 9.8% |
| + Both (Full) | **0.926** | **3.21** | **8.4%** |

*Baseline (MCMA-Net) corresponds to the original architecture [1], trained on our data split, without MSAGR or TAR-Loss.

Analysis of MSAGR design choices reveals that single-scale graphs ($8^3$ tokens) achieve 0.908 ET Dice at 7.2s inference, two-scale ($16^3$, $8^3$) improves to 0.916 at 7.8s, and three-scale ($32^3$, $16^3$, $8^3$) achieves 0.923 at 8.1s, while four-scale provides no additional benefit but increases inference time to 9.6s. Three scales optimally balance performance and efficiency, with coarse scales capturing global structure and finer scales modeling boundary

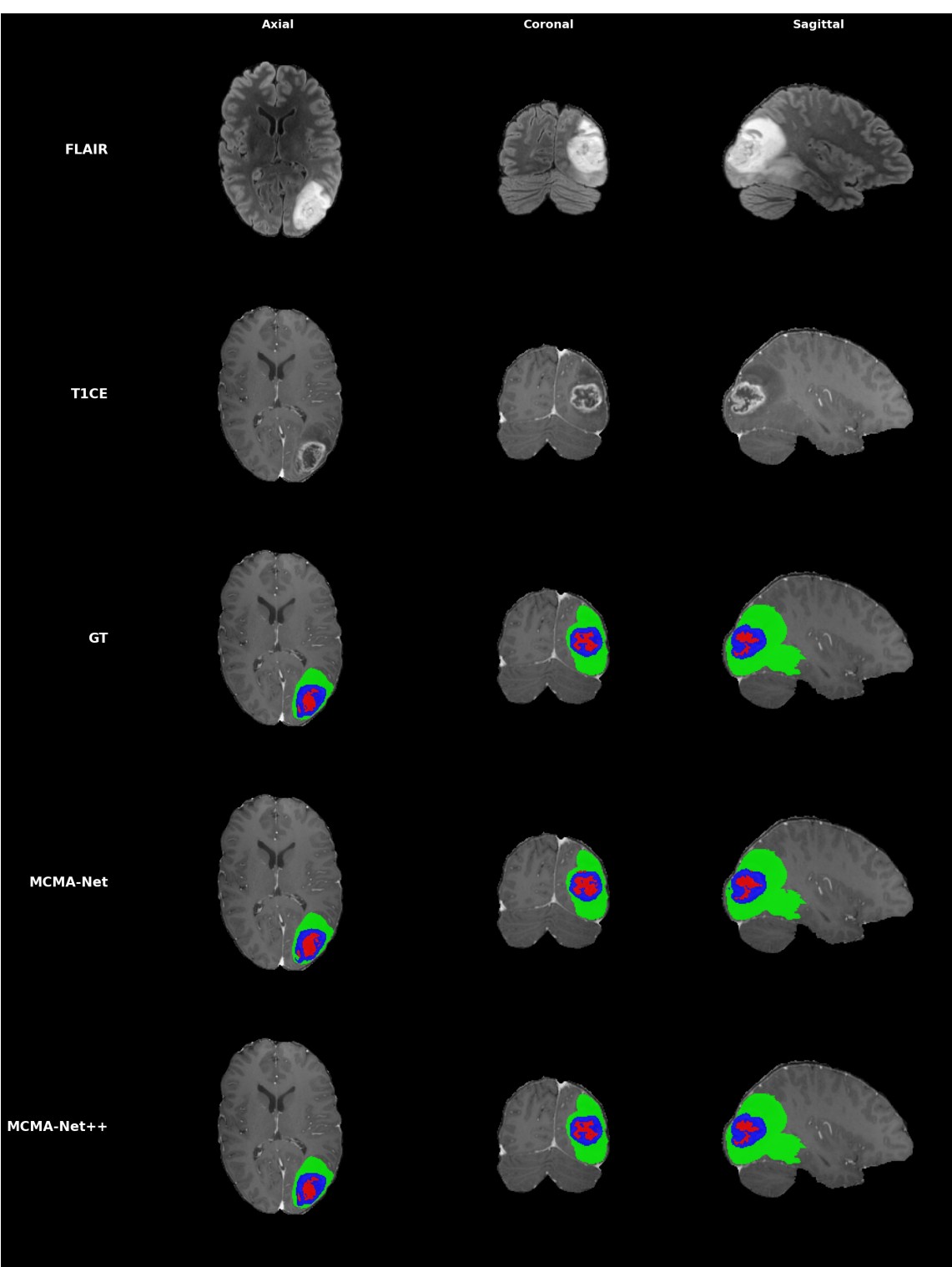

Figure 1: Qualitative comparison on BraTS 2021 case 00048 across three anatomical views. Rows show: FLAIR and T1ce modalities, GT annotations, MCMA-Net predictions, and MCMA-Net++ predictions. Color coding: green = WT, red = TC, blue = ET.

details. The three-scale configuration ($4^3$, $8^3$, $16^3$ patches) provides a good trade-off between representation capacity and efficiency, as these scales naturally correspond to fine-grained texture, intermediate tumor structures, and global anatomical context. Introducing an additional scale does not yield further gains, as it either exceeds meaningful anatomical context or overlaps with existing scales. This behavior is task-specific rather than a fundamental limitation. Memory usage scales linearly with input volume (e.g., 2.1 GB for $128^3$ and 16.8 GB for $256^3$), remaining feasible on standard clinical GPUs.

Incremental TAR-Loss analysis shows that from Dice loss only (0.900 ET Dice, 14.3% topology errors), adding $\mathcal{L}_{\text{inc}}$ provides the largest single-component gain (0.908, 11.8%), $\mathcal{L}_{\text{bnd}}$ further improves to (0.915, 10.1%), $\mathcal{L}_{\text{smooth}}$ reaches (0.921, 8.9%), and full TAR-Loss achieves (0.926, 8.4%). Each component contributes at different levels—voxel, boundary, surface, region—providing comprehensive topology enforcement.

Although topology violations occur in 8.4% of cases, their severity is low (0.9 voxels per case), with most confined to anatomically ambiguous boundaries known for high inter-rater variability. Similar error levels across HGG and LGG cases and ablation results indicate that remaining violations reflect clinical uncertainty rather than systematic failure, while confirming the complementary behavior of TAR-Loss and MSAGR.

## 6. Conclusion

In this work, we introduced MCMA-Net++, a topology-aware and graph-driven architecture for 3D glioma segmentation that substantially advances the state of the art in anatomically consistent tumor delineation. Through the synergistic combination MSAGR and TAR-Loss, our method achieves unprecedented performance on the BraTS 2021 benchmark: Dice scores of 0.970 (WT), 0.943 (TC), and 0.926 (ET), with HD95 of 3.21 mm and 41% reduction in topological violations compared to MCMA-Net.

From a clinical perspective, MCMA-Net++ addresses a critical bottleneck in glioma management. Manual tumor segmentation by expert neuroradiologists is extremely time-consuming (4-6 hours per patient) and subject to inter-observer variability, creating significant workflow constraints in busy radiology departments. MCMA-Net++ provides fully automated segmentation in under 10 seconds while maintaining anatomically plausible predictions that match or exceed inter-rater agreement levels. This dramatic efficiency gain enables radiologists to allocate more time to complex diagnostic decision-making, multidisciplinary tumor board discussions, and direct patient care. Moreover, the model's ability to maintain strict hierarchical constraints (ET $\subset$ TC $\subset$ WT) with only 8.4% topology violations ensures clinical trustworthiness for integration into treatment planning, surgical navigation, and radiotherapy workflows.

Looking forward, we plan to validate MCMA-Net++ on real-world clinical data from the University Hospital Center of Poitiers, where it will be evaluated on routine clinical MRI acquisitions with varying protocols and scanner characteristics. This prospective evaluation will assess the model's ability to generalize beyond the curated BraTS dataset and determine its practical value for radiologists, from initial diagnosis to longitudinal treatment monitoring. By bridging the gap between algorithmic innovation and clinical application, MCMA-Net++ has the potential to enhance diagnostic efficiency, improve treatment planning, and ultimately contribute to better outcomes for patients with glioma.

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
