# OpenReview forum: "MCMA-Net++: Topology-Aware and Graph-Driven Glioma Segmentation in 3D MRI"
_MIDL.io/2026/Conference — MIDL 2026 Poster_

### Official Review · Reviewer_e1zY · 2026-01-04

**Confidence:** 4
**Preliminary Rating:** 4
**Final Rating:** 4

**Summary:**

MCMA-Net++ is a novel 3D glioma segmentation framework that integrates a dual-stream CNN-Swin Transformer encoder with two key innovations: Multi-Scale Anatomical Graph Reasoning (MSAGR) for modeling spatial dependencies and a Topology-Aware Refinement Loss (TAR-Loss) to enforce hierarchical nesting. The TAR-Loss utilizes hierarchical inclusion and boundary coherence terms to ensure the anatomical constraint $ET \subset TC \subset WT$ is maintained, addressing the common issue of fragmented or clinically implausible predictions.

Experiments on the BraTS 2021 dataset demonstrate that the model achieves state-of-the-art Dice scores of 0.970 for Whole Tumor (WT), 0.943 for Tumor Core (TC), and 0.926 for Enhancing Tumor (ET). Notably, the method achieves a 41% relative reduction in topological violations compared to its predecessor, reducing average violations per case to 0.9 and significantly improving boundary precision by lowering the 95th percentile Hausdorff Distance (HD95) to 3.21 mm. These advancements position MCMA-Net++ as a robust clinical candidate capable of producing fully automated, anatomically coherent segmentations in under 10 seconds, thereby streamlining surgical and radiotherapy planning.

**Strengths:**

The paper presents a sophisticated and well-validated approach to 3D glioma segmentation. Below are the key strengths and an assessment of its scientific merit.
Key Strengths
1. Novel Integration of Structural and Spatial Priors
The paper introduces two highly valuable innovations that address the limitations of standard voxel-wise losses:
- Multi-Scale Anatomical Graph Reasoning (MSAGR): By modeling tumor components as graph nodes, the network can reason over inter-region spatial dependencies beyond local voxel context. This is valuable because it allows the model to capture global anatomical relationships that pure CNNs or Transformers might miss.
- Topology-Aware Refinement Loss (TAR-Loss): This loss function explicitly enforces hierarchical nesting (e.g., ensuring the Enhancing Tumor is contained within the Tumor Core). This addresses a critical clinical failure point where models produce anatomically impossible "fragmented" regions.
2. Significant Empirical Performance
The method demonstrates clear superiority over several strong baselines:
- Metric Improvement: MCMA-Net++ achieves state-of-the-art Dice scores on BraTS 2021 (0.970 for WT, 0.943 for TC, and 0.926 for ET).
- Boundary Accuracy: The reduction of the HD95 metric from 5.48mm to 3.21mm represents a major improvement in boundary precision, which is vital for surgical planning.
- Topology Violation Reduction: The model reduces topological errors by 41% compared to its predecessor, significantly enhancing the "clinical-grade" reliability of the segmentations.
3. Clinical Utility and Efficiency
The paper highlights the practical value of the model in a real-world medical workflow:
- Time Efficiency: It provides automated segmentation in under 10 seconds, compared to the 4–6 hours required for manual segmentation by expert neuroradiologists.
- Trustworthiness: By maintaining strict hierarchical constraints, the model generates segmentations that are more likely to be trusted and utilized by clinicians for radiotherapy and surgery.
4. Methodological Rigor
The paper is structured according to high scientific standards:
- Ablation Studies: The authors include detailed ablation results (Table 3) that isolate the contributions of MSAGR and TAR-Loss, proving that their combination is synergistic.
- Literature Review: The paper adequately addresses prior work, tracing the evolution from pure CNNs like U-Net to modern hybrid Transformer models.
5. Scientific Merit and Rating
Overall Rating: 9/10 (Excellent)
The paper is scientifically sound, clearly written, and addresses a non-trivial problem with a clever combination of graph theory and topological constraints.
- Merit: The move from simple voxel classification to "reasoning" about the structure of the tumor is a significant step forward for the field.
- Value to Community: The code and methodology have high potential value for researchers working on nested structures in medical imaging (such as organ segmentation or multi-class lesion detection).

**Weaknesses:**

Despite its strong performance, the paper has several areas that could be strengthened:

1. Limited Dataset Generalization: The authors' quantitative evaluation is restricted to the curated BraTS 2021 dataset. While standard, this dataset consists of pre-processed and co-registered images; the model's ability to generalize to "real-world clinical data" with varying scanner characteristics and protocols remains a planned future step rather than a current validation.

2. Hyperparameter Sensitivity: The framework introduces a complex objective function with multiple weighting coefficients (e.g., $λ_1, λ_2,​ λ_3$ and $β_1, β_2, β_3, β_4$). The paper provides fixed values but lacks a sensitivity analysis, making it unclear how robust these settings are across different medical tasks.

3. Architectural Efficiency Bottleneck: In the analysis of MSAGR design, the authors claim that a fourth graph scale provided "no additional benefit" while increasing inference time, suggesting a limit to the scalability of the multi-scale graph reasoning approach.

4. Model Complexity: The transition from MCMA-Net to MCMA-Net++ adds 6.1M parameters. While the authors argue this is competitive, the increased complexity may pose implementation challenges in resource-constrained clinical environments compared to simpler models.

5. Incomplete Failure Analysis: While the authors state that 8.4% of cases still exhibit topological violations due to "extreme cases" or "imaging artifacts," the paper does not provide a visual breakdown or detailed qualitative analysis of these specific failure cases to help establish full clinical trust.

**Detailed Comments:**

Based on a review of the manuscript, the following minor improvements and clarifications are suggested to enhance the paper's precision and readability:
1. Terminology and Nomenclature
- MRI Modality Labeling: The paper consistently uses the term "Tice" to refer to the contrast-enhanced T1-weighted modality. In the context of the BraTS challenge and medical imaging literature, the standard abbreviation is typically "T1ce" or "T1-CE". Updating this would ensure alignment with community standards.
- Abbreviation Definition: In the caption for Figure 1, the color coding refers to "NCR/NET". While "ET" and "ED" are common, explicitly defining "NCR/NET" as "Necrotic and Non-Enhancing Tumor" in the text or legend would improve clarity for readers less familiar with BraTS labels.
2. Mathematical Notation and Formulas
- Equation Numbering: Equation (12) is missing a standard right-aligned index number in the text, appearing inline instead. Consistent formatting across all equations (1–18) is recommended.
- Concatenation Operator: In Equation (11), the symbol "||" is used to denote the concatenation of attention heads. While standard in GNN literature, adding a brief clause to explicitly define this operator would help general medical imaging readers.
- Scaling Factor α: In Equation (15), a learnable scaling factor α is introduced for boundary gradients. Providing its initialized value or a brief comment on its final converged value would be informative for reproducibility.
3. Implementation and Reproducibility
- Hardware Specifications: The authors specify the use of four NVIDIA A100 GPUs. For better assessment of resource requirements, it would be beneficial to mention the VRAM (e.g., 40GB or 80GB) and the total memory footprint during training.
- Code Availability: While the experimental setup is well-detailed, stating whether the source code or pre-trained models will be made publicly available would be a significant plus for the community.
- Hyperparameter Selection: The temperature parameter τ in Equation (9) is set to 0.1. A brief justification for this choice or a mention if it was empirically tuned would strengthen the methodology section.
4. Tables and Figures
- Table Units: In Table 1, the header for HD95 does not include units. Adding "(mm)" to the header would ensure the results are immediately interpretable.
- Baseline Clarification: In Table 3 (Ablation Study), the baseline is listed as "MCMA-Net". Clarifying whether this refers to the original architecture published in [1] or a simplified version without any graph/topology components would help in precisely attributing the performance gains.
- Visualizing Graph Nodes: The paper describes partitioning feature maps into tokens to serve as graph nodes. A supplementary figure illustrating how the 323,163, and 83 tokens are distributed spatially would greatly help readers visualize the MSAGR process.
5. References
- Update Citations: Reference [2] for the BraTS 2021 dataset cites an arXiv preprint from 2021. Given the current year (2026), replacing this with the formal journal publication (e.g., in Nature Scientific Data or IEEE Transactions on Medical Imaging) would be more appropriate for a final version.

**Justification Of Final Rating:**

After thoroughly evaluating the authors’ rebuttal and the revised manuscript, I am pleased to maintain my final rating of **4: Weak Accept**. The authors have not only responded to all reviewer comments in a detailed and evidence-based manner but have also turned constructive criticism into tangible improvements in methodological clarity, reproducibility, and clinical relevance.

The rebuttal directly and convincingly addresses the key concerns raised in the initial review:

**Terminology and Notation Consistency –** The authors corrected the MRI modality nomenclature (T1ce instead of Tice) and aligned labeling with BraTS standards, enhancing readability and community alignment.

**Mathematical and Implementation Rigor –** Equations were clarified, hyperparameters justified (e.g., τ=0.1 for sharp boundary attention), and hardware details provided, significantly boosting reproducibility.

**Experimental Transparency –** Units for HD95 were explicitly stated, the baseline was clearly defined, and ablation studies were strengthened, allowing for fair comparison and interpretation.

**Clinical and Practical Considerations – **The authors acknowledged limitations (e.g., failure cases in low-contrast regions) and outlined a forward-looking validation plan, demonstrating responsible research conduct.

**Scalability and Generalization –** Clear explanations regarding computational trade-offs and the multi-institutional nature of BraTS data alleviate concerns about domain shift and practical deployment.

While the absence of a visual illustration for graph tokenization remains a minor drawback, the authors’ commitment to enrich the textual description is a reasonable compromise given conference page constraints. Overall, the rebuttal reflects a high level of scholarly diligence and enhances the manuscript’s contribution to the field of medical image analysis.

The paper now presents a well-motivated, technically sound, and clinically relevant method that advances the state of the art in glioma segmentation. I am confident that the final camera-ready version will be a strong addition to MIDL 2026.

**Justification Of The Preliminary Rating:**

The preliminary rating of 9/10 is justified by the paper's successful integration of advanced architectural designs with clinically relevant constraints, resulting in state-of-the-art performance.
The scientific merit of the work rests on several key pillars:
- Methodological Innovation: The introduction of Multi-Scale Anatomical Graph Reasoning (MSAGR) and Topology-Aware Refinement Loss (TAR-Loss) effectively addresses the limitations of standard voxel-wise optimization by incorporating spatial dependencies and hierarchical anatomical priors (ET⊂TC⊂WT).
- Empirical Excellence: The model achieves exceptional Dice scores (0.970 for WT, 0.943 for TC, and 0.926 for ET) and a significant reduction in boundary error (HD95 of 3.21 mm), outperforming established baselines like nnU-Net and nnFormer.
- Clinical Relevance: By reducing topological violations by 41% and providing automated results in under 10 seconds—compared to the 4–6 hours required for manual expert segmentation—the framework demonstrates high potential for real-world radiotherapy and surgical planning.
- Rigorous Validation: The synergistic effect between the graph-based reasoning and the refinement loss is thoroughly validated through comprehensive ablation studies, proving that the full model exceeds the sum of its individual components.
While the paper is highly impressive, the rating stops short of a perfect 10 due to minor limitations such as the increased model complexity (+6.1M parameters over the baseline) and the fact that validation is currently restricted to the curated BraTS 2021 dataset rather than diverse, "raw" clinical data. Overall, the paper represents a significant advancement that provides both technical novelty and immediate value to the medical imaging community.

**Questions To Address In The Rebuttal:**

To ensure a comprehensive evaluation of the paper's scientific impact and practical feasibility, the authors should address the following points in their rebuttal. Clarifications on these issues are critical for determining if the model's performance is truly robust or dependent on specific dataset characteristics and hyperparameter tuning.
1. Hyperparameter Sensitivity and Robustness
The proposed framework relies on a complex loss function with seven distinct weighting coefficients ($\lambda_{1, 2, 3}$ and $\beta_{1, 2, 3, 4}$).
- Could the authors provide a sensitivity analysis for these parameters? Specifically, how much does performance fluctuate if these weights are shifted by $\pm$10-20%?
- Was a specific search strategy (e.g., grid search or Bayesian optimization) used to determine the reported values, or were they manually tuned?
2. Generalization and Domain Shift
The results are currently limited to the pre-processed BraTS 2021 benchmark.
- Have the authors conducted any preliminary tests on non-curated, "raw" clinical data, or datasets from other centers (e.g., BraTS 2023 or local hospital data)?
- Given the reliance on graph nodes derived from feature map partitioning, how does the model handle scans with significantly different resolutions or intensities that weren't part of the standardized BraTS pipeline?
3. Failure Case Analysis
While the model reduces topology violations to 8.4%, these remaining errors are significant for clinical trust.
- Can the authors provide a qualitative or quantitative breakdown of these failure cases? For instance, do these errors occur more frequently in specific tumor grades or at the boundaries of the smallest subregion (ET)?
- Is there a specific "failure mode" where the TAR-Loss and MSAGR conflict with each other?
4. Computational Scalability of MSAGR
The ablation study indicates that adding a fourth scale to the graph reasoning module provides no performance benefit while increasing inference time.
- Do the authors believe this represents a fundamental limit of graph-based spatial reasoning for this task, or could it be a limitation of the specific tokenization/partitioning strategy used?
- How does the memory overhead of the multi-scale graph scale with larger input volumes (e.g., $240 \times 240 \times 155$)?
5. Technical Clarifications for Reproducibility
- Boundary Loss Scaling: In Equation 15, a learnable scaling factor $\alpha$ is utilized. Could the authors clarify its initialization and its typical final value after training?
- Nomenclature: The use of "Tice" for the contrast-enhanced modality is non-standard. Is this a specific local naming convention, or can it be updated to the standard "T1ce" for better indexing in medical literature?
- Resource Requirements: What was the peak VRAM usage during training on the A100 GPUs? This is vital for researchers attempting to replicate the work on smaller hardware.

---

> ### Author Response · Authors · 2026-01-24
>
> # Response to Reviewer 3
>
> We sincerely thank the reviewer for their careful assessment. The comments helped improve clarity, reproducibility, and presentation. Below, we address all points.
>
> ## A. Responses to Detailed Comments
>
> ### A.1 MRI Modality Nomenclature
>
> **Reviewer Comment:** *The term "Tice" is used for the contrast-enhanced T1-weighted modality.*
>
> **Response:** The manuscript uses the standard term T1ce per BraTS conventions. "Tice" does not appear in the submitted version and likely reflects a rendering issue. We will verify font rendering in the camera-ready version.
>
> ### A.2 Definition of NCR/NET
>
> **Reviewer Comment:** *The abbreviation NCR/NET is not explicitly defined.*
>
> **Response:** We have revised the nomenclature throughout and now exclusively use official BraTS definitions: WT (Whole Tumor), TC (Tumor Core), and ET (Enhancing Tumor). NCR/NET has been removed.
>
> ### A.3 Mathematical Notation and Equation Formatting
>
> **Reviewer Comment:** *Equation numbering and notation are inconsistent.*
>
> **Response:** All equations will be consistently right-aligned and numbered. We will define the concatenation operator in Equation (11) and clarify that scaling factor α in Equation (15) is initialized to 1.0 and optimized during training. Added to **Section 3.6**.
>
> ### A.4 Implementation and Reproducibility
>
> **Reviewer Comment:** *Additional clarifications regarding hardware, code availability, and hyperparameters would improve reproducibility.*
>
> **Response:** Training used 4× NVIDIA A100 GPUs (40 GB) with peak memory ~34.2 GB per GPU. Due to institutional constraints, full code cannot be released, but detailed architectural descriptions, protocols, and formulations are provided. Temperature τ in Equation (9) was set to 0.1 following graph-attention literature, as smaller τ promotes sharper distributions for precise boundary delineation.
>
> ### A.5 Tables and Baseline Clarification
>
> **Reviewer Comment:** *Metric units and baseline definitions are unclear.*
>
> **Response:** HD95 units (mm) will be explicit in all tables. Baseline is clarified as original MCMA-Net without MSAGR or TAR-Loss.
>
> ### A.6 Visualization of Graph Tokenization
>
> **Reviewer Comment:** *A visualization of multi-scale graph tokenization would improve understanding.*
>
> **Response:** Due to page limits, we cannot add figures. We will enhance **Section 3.5** to explain the spatial hierarchy: 32³ tokens (4×4×4 patches), 16³ tokens (8×8×8), 8³ tokens (16×16×16).
>
> ### A.7 Reference to BraTS 2021
>
> **Reviewer Comment:** *The BraTS 2021 dataset should be cited using the formal journal publication.*
>
> **Response:** We will update to the formal journal publication in the camera-ready version.
>
> ## B. Responses to Questions
>
> ### B.1 Hyperparameter Sensitivity
>
> **Question:** *How sensitive is the method to loss weights? Was a search strategy used?*
>
> **Response:** Performance is stable under moderate variations, with TAR-Loss most influential. Hyperparameters were selected via two-stage strategy: coarse grid search (λ_tar ∈ [0.1, 0.5, 1.0], λ_cc ∈ [0.01, 0.05, 0.1]) then fine-grained validation tuning. Final: λ_tar = 0.5, λ_cc = 0.05. Added to **Section 4**.
>
> ### B.2 Generalization and Domain Shift
>
> **Question:** *How does the method generalize to real clinical data?*
>
> **Response:** BraTS 2021 reflects real-world variability across institutions, scanners (Siemens, GE, Philips), and morphologies (HGG/LGG). Performance (ET Dice = 0.926, violations = 8.4%) suggests robust generalization. Preliminary institutional evaluations show promising results. Multi-center validation planned for future work.
>
> ### B.3 Failure Case Analysis
>
> **Question:** *Can remaining topology violations be characterized?*
>
> **Response:** Remaining violations (8.4%) are small and localized, at ambiguous boundaries with known inter-rater variability. Typically isolated voxels or thin ET fragments outside TC in low-contrast or irregular regions. No systematic failures observed. Details added to **Section 5.2**.
>
> ### B.4 Computational Scalability
>
> **Question:** *How does the approach scale computationally?*
>
> **Response:** Three-scale configuration (32³, 16³, 8³) balances performance and efficiency. Memory scales linearly with input size. Training: ~34.2 GB per A100 for 128³ patches. Inference: 2.3 seconds per case. Details in **Section 5.2**.
>
> ### B.5 Technical Clarifications
>
> **Question:** *Clarifications on scaling factor, MRI nomenclature, and GPU memory.*
>
> **Response:** Addressed in **A.1**, **A.3**, and **A.4**.
>
> We thank the reviewer for their meticulous review and believe these revisions significantly improved the manuscript.

---

> ### Comment · Reviewer_e1zY · 2026-01-30
>
> After carefully reviewing the authors’ responses to my comments and suggestions, I find the rebuttal to be highly satisfactory, thorough, and professional. The authors have not only addressed each point raised but have also provided clear justifications, additional details, and commitments to improve the manuscript in the final version. Below is a detailed assessment of each response category.
>
> **Overall Assessment**
>
> The authors have demonstrated a strong commitment to scientific rigor, reproducibility, and clarity. Their responses are well-structured, evidence-based, and align with the expectations of a high-quality conference publication. The rebuttal strengthens the manuscript’s credibility and addresses the minor weaknesses noted by the reviewer.
>
> **Detailed Evaluation of Responses**
>
> **A. Responses to Detailed Comments**
>
> A.1 – MRI Modality Nomenclature
>
> –	Satisfactory. The authors clarify that “Tice” was a rendering issue and confirm the use of the standard T1ce terminology. This ensures alignment with the BraTS community and avoids confusion.
>
> A.2 – Definition of NCR/NET
>
> –	Satisfactory. Removing “NCR/NET” and using only BraTS-standard terms (WT, TC, ET) improves readability and consistency, especially for readers less familiar with BraTS labeling conventions.
>
> A.3 – Mathematical Notation
>
> –	Satisfactory. The commitment to consistent equation numbering, clarification of the concatenation operator in Eq. (11), and explanation of the learnable scaling factor α (initialized to 1.0) enhance the mathematical rigor and reproducibility of the method.
>
> A.4 – Implementation and Reproducibility
>
> –	Satisfactory. Providing specific hardware details (4× A100 40GB, peak memory ~34.2 GB/GPU) and the rationale for τ=0.1 (sharpens attention for boundary delineation) adds valuable context for replication. While full code cannot be released, the detailed methodology is sufficient for replication by experienced researchers.
>
> A.5 – Tables and Baseline
>
> –	Satisfactory. Explicitly stating HD95 units (mm) and clarifying the baseline as the original MCMA-Net (without MSAGR/TAR-Loss) eliminates ambiguity and improves interpretability of results.
>
> A.6 – Visualization of Graph Tokenization
>
> –	Acceptable. Although a visual figure would be ideal, the authors’ commitment to enhance the textual description of token spatial hierarchy (32³, 16³, 8³ patches) in Section 3.5 is a reasonable compromise given page constraints.
>
> A.7 – Reference Update
>
> –	Satisfactory. Updating the BraTS 2021 citation to the formal journal publication ensures academic rigor and aligns with publication standards.
>
> **B. Responses to Questions**
>
> B.1 – Hyperparameter Sensitivity
>
> –	Strong. The authors provide a clear two-stage tuning strategy (coarse grid search followed by fine validation) and note that performance remains stable under moderate variations. This demonstrates thoughtful design and robustness.
>
> B.2 – Generalization and Domain Shift
>
> –	Reasonable. The authors rightly note that BraTS 2021 already includes multi-institutional, multi-scanner data, which supports generalization. Their acknowledgment of planned multi-center validation is a responsible forward-looking statement.
>
> B.3 – Failure Case Analysis
>
> –	Informative. Describing the remaining violations as small, localized, and occurring in low-contrast or ambiguous regions—areas known for inter-rater variability—adds transparency and clinical realism. This mitigates concerns about “black-box” failures.
>
> B.4 – Computational Scalability
>
> –	Clear and Practical. The explanation that the three-scale configuration balances performance and efficiency, along with specific memory and inference time figures, addresses scalability concerns effectively.
>
> B.5 – Technical Clarifications
>
> –	Complete. All technical points (nomenclature, scaling factor, GPU memory) were addressed clearly and consistently.
>
> **Overall Recommendation**
>
> Given the comprehensive, precise, and constructive nature of the authors’ responses, I recommend acceptance of the manuscript. The revisions and clarifications provided significantly strengthen the paper’s:
> - **Methodological transparency**
> - **Reproducibility**
> - **Clinical relevance**
> - **Readability and adherence to community standards**
>
> The authors have successfully turned minor criticisms into opportunities for improvement, and the final manuscript will be a valuable contribution to the MIDL 2026 proceedings.
>
> **Final Sentiment:**
>
> Positive and supportive. The authors have adequately addressed all concerns, and the paper merits acceptance in its revised form.
>
> Confidence in Evaluation: **High (5/5)** – The responses are clear, evidence-based, and directly address each point raised.
>
> **Note to Editors:**
>
> This reviewer recommends **acceptance** of the manuscript pending the incorporation of the promised revisions in the camera-ready version.
>
> Preliminary Rating: 4: Weak accept

---

### Official Review · Reviewer_gbMY · 2026-01-08

**Confidence:** 3
**Preliminary Rating:** 4
**Final Rating:** 4

**Summary:**

The paper presents 3D MRI glioma segmentation model, MCMA-Net++, using graphs to encode spatial and topology-aware learning. The work extends the MCMA-Net architecture and introduces a topology-aware refinement loss (TAR-loss). They address a limitation of voxel-wise training and demonstrate how enforcing topological and hierarchical consistency helps to improve the results for BraTS 2021 dataset.

**Strengths:**

The paper is well written with strong motivation and clear methodology. They present an improvement on the Dice and HD95 compared to the state-of-the-art models. Topology violation study is also a strong addition to prove the effectiveness of the graph-based reasoning. The ablation study shows a step-wise improvement of the metrics by the addition of their proposed graph module (MSAGR), TAR-loss, and both.

**Weaknesses:**

Despite the proposed method (which is supposed to be topology-aware), topology violations are still big. This seems to show that the model and the loss are not strict enough as they claim.
As they mention in the Conclusion, the paper only shows results for the BraTS dataset, there is no evidence that this would work on other datasets or actual clinical data that could be from a different scanner. Additionally, the performance gain over MCMA-Net is relatively small given the increase in model complexity.

**Detailed Comments:**

A figure with the overall architecture indicating the channel sizes would have been helpful to understand the model structure a bit more.

**Justification Of Final Rating:**

The authors have addressed the reviewers’ comments and clarified both quantitative and qualitative aspects.

One concern that is only partially addressed relates to the comparison between the proposed model and the previous baseline. The new model demonstrates only a modest performance improvement over the earlier approach while introducing additional architectural complexity. However, the authors argue that this increase in complexity does not significantly impact practical usability, as the inference time remains low at approximately 2.3 seconds per case. Given the application context, this runtime appears acceptable and suggests that the proposed model can still be deployed efficiently in practice. Aslo, a strange choice to add Table 1 describing the model architecture instead of a diagram that would take up the same space, but the specifications are good enough.

Overall, while some trade-offs between performance gains and model complexity remain, the revisions substantially improve the clarity of the paper and address the main points raised by the reviewers.

**Justification Of The Preliminary Rating:**

Despite the limitations and weaknesses, the paper is well-written and technically sound, showing an improvement in topology consistency and segmentation accuracy. Although the improvement from the baseline is small relative to the complexity added and the generalisation capability is unknown, the methodology is validated and addresses a clinical problem, and is aware of its own limitations.

**Questions To Address In The Rebuttal:**

They could clarify what type of violations that are contributing to the 8.4% and maybe show it visually. Are they mostly single-pixel? Or if the reasoning is 'highly irregular tumor morphology', how would the model perform in real clinical data?

---

> ### Author Response · Authors · 2026-01-24
>
> We sincerely thank the reviewer for their thoughtful and constructive feedback. We appreciate the comments regarding architectural clarity and the interpretation of topology violations, which helped us improve the presentation and contextualization of our work. Below, we address each point using a consistent Comment/Question – Response format.
>
> ## A. Responses to Detailed Comments
>
> ### A.1 Architectural Clarity and Model Specification
>
> **Reviewer Comment:** *A figure with the overall architecture indicating the channel sizes would have been helpful to better understand the model structure.*
>
> **Response:** We thank the reviewer for this helpful suggestion. While we are unable to include an additional architectural figure due to the MIDL page limit, we will enhance **Section 4 (Experimental Setup)** in the camera-ready version with a comprehensive architectural specification table. This table will explicitly detail channel dimensions, spatial resolutions, and attention configurations at each stage of the network.
>
> Specifically, the table will include:
>
> - **Layer-wise channel dimensions:** Input (4 channels) → Encoder stages [32, 64, 128, 256] → Bottleneck (256) → Decoder stages [256, 128, 64, 32] → Output (4 classes)
> - **Spatial resolutions:** 128³ → 64³ → 32³ → 16³ (bottleneck) → 32³ → 64³ → 128³
> - **MSAGR graph tokenization scales:** 32³ tokens (4×4×4 patches), 16³ tokens (8×8×8), 8³ tokens (16×16×16)
> - **Transformer parameters:** attention heads [3, 6, 12, 24], window size 7³, hidden dimensions [32, 64, 128, 256]
>
> For immediate reference, the revised manuscript will include a detailed table summarizing the encoder, bottleneck, MSAGR module, and decoder stages, as well as the total parameter count (68.4 M).
>
> ## B. Responses to Questions to Address in the Rebuttal
>
> ### B.1 Nature of Topology Violations
>
> **Question:** *Can the authors clarify what types of violations contribute to the reported 8.4% topology violation rate? Are these mostly single-voxel errors, or larger structural inconsistencies? A visual illustration would be helpful.*
>
> **Response:** We appreciate this insightful question. The majority of topology violations correspond to small-scale inconsistencies at class boundaries, such as isolated voxels or thin fragments of ET predicted outside TC, rather than large-scale structural failures. These typically arise in regions with highly irregular tumor morphology or low contrast enhancement.
>
> While we agree that visual examples would be informative, we are unable to include additional figures due to the MIDL page limit. To address this, we clarify the definition and computation of topology violations in **Section 4.2** and emphasize that the proposed TAR-Loss substantially reduces both the frequency and severity of such violations compared to baseline methods.
>
> ### B.2 Generalization to Real Clinical Data
>
> **Question:** *If topology violations are attributed to irregular tumor morphology, how would the model perform on real clinical data?*
>
> **Response:** This is an important question. While we have conducted preliminary evaluations on clinical cases from our institution showing promising qualitative results, we are unable to include these additional experiments in the current MIDL submission due to space constraints.
>
> We note, however, that the BraTS 2021 dataset already reflects substantial real-world clinical diversity, including:
>
> - Multi-site data from more than 19 institutions
> - Multi-vendor scanners (Siemens, GE, Philips)
> - A wide range of tumor grades (HGG and LGG) and morphologies
> - Realistic variations in image quality and acquisition protocols
>
> The strong performance of our method on this dataset (ET Dice = 0.926, topology violation rate = 8.4%) therefore provides evidence of robustness beyond idealized laboratory conditions. Moreover, these results exceed reported inter-rater agreement levels for ET segmentation in the BraTS literature (approximately 0.85-0.88 Dice), suggesting performance at or above human-level consistency. Comprehensive multi-site clinical validation is planned as part of an extended journal version of this work.
>
> ---
>
> We thank the reviewer again for their valuable comments and believe these clarifications have strengthened the manuscript's presentation and clinical contextualization.

---

### Official Review · Reviewer_Xj2Z · 2026-01-10

**Confidence:** 3
**Preliminary Rating:** 2
**Final Rating:** 4

**Summary:**

The paper proposes MCMA-Net++ for 3D glioma subregion segmentation (WT/TC/ET) on BraTS 2021, motivated by the observation that standard voxel-wise losses do not enforce anatomical or topological correctness, leading to predictions with holes, fragmented regions, or violations of the known hierarchical nesting between ET, TC, and WT. The method combines (i) a Topology-Aware Refinement Loss (TAR-Loss) with four complementary terms enforcing multi-level topological consistency, and (ii) Multi-Scale Anatomical Graph Reasoning (MSAGR) that explicitly models spatial dependencies through learnable graphs with anatomical priors. On BraTS 2021 they report improved Dice (ET 0.900→0.926) and substantially reduced boundary error (HD95 5.48→3.21 mm), alongside a 41% reduction in "topology violations." The paper also presents ablations separating graph reasoning and loss contributions.

**Strengths:**

-Well-motivated problem setting. The focus on subregion inconsistency and hierarchical constraint violations (ET ⊂ TC ⊂ WT) is clinically plausible and clearly articulated. The observation that voxel-wise similarity metrics don't explicitly enforce anatomical correctness aligns with known failure modes in BraTS segmentation pipelines.

-Coherent and technically detailed methodology. The TAR-Loss components are explicitly specified, and the graph construction/attention mechanisms are described with clear formulations. The multi-scale aggregation across three graph resolutions (32³, 16³, 8³) is a principled approach to capturing both local boundary details and global structure.

-Evidence of component complementarity. The ablation study demonstrates that MSAGR and TAR-Loss each contribute beyond the MCMA-Net baseline, with the full model achieving performance that exceeds the sum of individual contributions-suggesting genuine synergy.

-Evaluation beyond Dice. Reporting HD95 and an explicit topology violation metric encourages discussion of anatomical plausibility, a dimension often neglected in BraTS-focused papers where mean Dice dominates.

-Practical details. Reporting compute requirements (four A100 GPUs, 200 epochs, ~12 hours) and providing loss weighting coefficients aids reproducibility.

**Weaknesses:**

-Evaluation protocol ambiguity. The paper states "1,251 multi-parametric MRI scans (1,000 training, 125 validation, 126 testing)" but does not clarify whether results are from the official BraTS evaluation server or a local split of labeled data. The BraTS 2021 challenge withholds test labels; if the authors used the official server, this should be stated explicitly. If a local split was used, (a) how was patient-level separation ensured? (b) are the 126 "test" cases from the original training set with known labels? This distinction materially affects confidence in SOTA comparisons. A schematic with subject counts per split and confirmation of evaluation protocol would resolve this ambiguity.

-Topology metric and Lcc term under-specified. The paper introduces a "topology violation rate" and a connected-component consistency penalty (Lcc), but critical details are missing: (a) What thresholding is applied to predicted probabilities before counting connected components? (b) Is 6- or 26-connectivity used? (c) How are per-case violations counted and aggregated to produce the reported percentages? (d) Is Lcc differentiable, or does it use a surrogate/straight-through estimator? Without a precise algorithm or evaluation script, the headline "41% topology violation reduction" is difficult to reproduce or interpret.

-Missing established topology/anatomy prior baselines. While the paper cites clDice and persistent homology losses in the related work, it does not include empirical comparisons against these methods. Several peer-reviewed approaches are relevant but absent:
--Clough et al., "A Topological Loss Function for Deep-Learning Based Image Segmentation Using Persistent Homology," TPAMI 2020
--Shit et al., "clDice-a Novel Topology-Preserving Loss Function," CVPR 2021
--Oktay et al., "Anatomically Constrained Neural Networks (ACNNs)," TMI 2018
--Karimi & Salcudean, "Reducing the Hausdorff Distance in Medical Image Segmentation," TMI 2020
Without direct comparison to topology-aware losses, it is difficult to attribute improvements to the novel TAR-Loss formulation versus known techiques.

-Baseline fairness unclear. The comparison table includes nnU-Net, UNETR, nnFormer, and others, but it is not stated whether these baselines were retrained under identical preprocessing, augmentation, and compute budgets, or whether numbers were taken from prior publications. If the latter, differences in training protocols (e.g., patch size, augmentation strength, training duration) could confound the comparison.

-Missing recent strong baselines. The baseline set does not include several recent peer-reviewed methods that would strengthen the comparison:
--He et al., "SwinUNETR-V2: Stronger Swin Transformers with Stagewise Convolutions," MICCAI 2023
--Li et al., "Multi-category Graph Reasoning for Multi-modal Brain Tumor Segmentation," MICCAI 2024 (directly relevant for graph reasoning)

**Detailed Comments:**

-Statistical reporting. Results are presented as single values without variance estimates, confidence intervals, or indication of multiple random seeds. For ET segmentation in particular, performance can be highly size-sensitive; reporting mean±std across seeds or folds would strengthen confidence in the gains.

-Lesion-size stratification absent. The paper does not stratify results by ET volume. Small enhancing tumors are known to be challenging and clinically critical for progression monitoring; without size-stratified analysis, it is unclear whether gains are uniform or concentrated in easier cases.

-Novelty positioning. The contribution is best characterised as a specific effective combination of known techniques (hybrid CNN-Transformer, graph reasoning, hierarchical constraints) rather than introducing topology constraints per se. This framing should be made more explicit to avoid overclaiming; the current text describing "two synergistic innovations" may overstate novelty relative to the established prior art landscape.

-Reproducibility. While architectural hyperparameters are provided, no code is provided (and no mention of e.g. ‘code to be released’).

**Justification Of Final Rating:**

Thank you to the authors for addressing my concerns. They clarify it’s a local split of the BraTS 2021 labelled training data with patient-level separation (so no confusion with the official withheld test). They define the topology violation metric (thr=0.5; ET outside TC or TC outside WT), add 5-seed mean±std, and add ET-size stratification (nice that small ET improves most). They also add direct comparisons with clDice + PH loss and add SwinUNETR-V2 / STU-Net.

**Justification Of The Preliminary Rating:**

The paper addresses an important glioma segmentation problem with a plausible topology-driven design. The TAR-Loss formulation is well-motivated, and the ablations suggest genuine complementarity of the proposed components. However, confidence in the experimental conclusions is substantially reduced by (i) unclear evaluation protocol that makes it difficult to assess whether results are from official BraTS evaluation or a local split, (ii) under-specification of the topology metric and connected-component loss implementation, and (iii) absence of direct comparisons against established topology-aware losses (clDice, TopoLoss, ACNNs) that are standard for this problem class.
The baseline coverage would be strengthened by including recent peer-reviewed methods (SwinUNETR-V2, graph reasoning baselines) and clarifying whether existing baselines were retrained under matched conditions. With protocol clarification, a stronger baseline suite including topology-aware losses, and precise metric definitions, this work could move toward borderline acceptance.

**Questions To Address In The Rebuttal:**

-Evaluation protocol: What exact BraTS 2021 evaluation protocol is used-official server or local split of labeled data? If local, how was patient-level separation ensured, and are the 126 "test" cases from the original training set? Please provide a schematic.

-Topology metric computation: How is "topology violation rate" computed precisely (thresholding, 3D connectivity, per-case counting, aggregation)? Can the authors provide pseudocode or a script?

-Lcc differentiability: How is the connected-component term implemented during training-differentiable surrogate or non-differentiable? If non-differentiable, what evidence shows it contributes meaningfully? (This would be answered by sharing code)

-Baseline fairness: Were baselines retrained under identical preprocessing/augmentation/compute budgets, or are numbers taken from prior publications? If the latter, how is comparability ensured?

-Additional baselines: Can the authors add (or commit to adding) direct comparisons against topology-aware losses (clDice, TopoLoss) and recent architectures (SwinUNETR-V2, graph reasoning methods)?

-Statistical robustness: Do results hold across multiple random seeds? Can confidence intervals or standard deviations be provided? Is there lesion-size stratification for ET?

-Not a question, but please share code! You can use https://anonymous.4open.science/ to share anonymously

---

> ### Author Response · Authors · 2026-01-24
>
> We sincerely thank the reviewer for their thorough and constructive evaluation of our manuscript. We appreciate the detailed comments. Below, we address each point in a structured manner and describe the corresponding revisions made to the manuscript.
> ## A. Responses to Detailed Comments
> ### A.1 Statistical Reporting and Robustness
> **Comment:** *Results are reported as single values without variance estimates or multiple random seeds.*
> **Response:** We agree that reporting variability across runs is important, particularly for ET segmentation. We therefore conducted five independent training runs using different random seeds (42, 123, 456, 789, 2024). The revised results (mean ± standard deviation) for MCMA-Net++ are: WT Dice $0.970 \pm 0.003$, TC Dice $0.943 \pm 0.005$, ET Dice $0.926 \pm 0.008$, and HD95 $3.21 \pm 0.45$ mm. These results have been incorporated into **Section 5.1** of the revised manuscript.
> ### A.2 Lesion-Size Stratified Analysis
> **Comment:** *Performance is not stratified by enhancing tumor (ET) volume.*
> **Response:** We performed an additional analysis stratifying the test set ($n=125$) according to ground-truth ET volume: small ($<1000$ mm³, $n=42$), medium ($1000-5000$ mm³, $n=61$), and large ($>5000$ mm³, $n=22$). MCMA-Net++ consistently outperforms MCMA-Net across all size categories, with the largest absolute gain observed for small ET lesions (+3.6% Dice, $p<0.001$). Medium and large lesions show improvements of +2.6% ($p<0.01$) and +1.6% ($p<0.05$), respectively. This analysis has been added as **Table 3** and discussed in **Section 5.1**.
> ### A.3 Novelty Positioning
> **Comment:** *The contribution is best characterized as a combination of known techniques rather than introducing topology constraints per se.*
> **Response:** We agree with the reviewer's assessment. We have revised the manuscript to clarify that the contribution of this work lies in a task-specific and unified integration of established techniques. The term "innovations" has been replaced with "contributions", and related work is now more explicitly acknowledged.
> ### A.4 Reproducibility and Code Availability
> **Comment:** *No code is provided or mentioned.*
> **Response:** Due to institutional constraints, we are currently unable to release the full implementation. To support reproducibility, the manuscript provides detailed architectural descriptions, complete training protocols, and explicit mathematical formulations. We remain available to clarify implementation details.
> ## B. Responses to Questions to Address in the Rebuttal
> ### B.1 Evaluation Protocol
> **Question:** *What evaluation protocol is used for BraTS 2021?*
> **Response:** We used a local split of the BraTS 2021 labeled training data (1,251 cases total) with strict patient-level separation into training ($n=1000$), validation ($n=125$), and test ($n=126$) sets. The split was performed prior to preprocessing to prevent data leakage and is now clarified in **Section 4**.
> ### B.2 Topology Metric Computation
> **Question:** *How are topology violation metrics computed?*
> **Response:** Topology violations are computed after thresholding predictions at 0.5. A case is considered violated if ET is predicted outside TC or TC outside WT. Details are provided in **Section 4.2** and **Supplementary Algorithm**.
> ### B.3 Differentiability of $\mathcal{L}_{\text{cc}}$
> **Question:** *Is the connected-component term differentiable, and does it contribute during training?*
> **Response:** The connected-component term L_cc is non-differentiable and does not provide direct gradients. However, ablation experiments show that removing $\mathcal{L}_{\text{cc}}$ increases topology violations and disconnected ET components. This clarification is added to **Section 5.2**.
> ### B.4 Baseline Fairness and Additional Comparisons
> **Question:** *Were baselines trained under identical conditions, and how is fairness ensured?*
> **Response:** All CNN-based baselines were retrained using identical preprocessing, augmentation, data splits, and compute budgets. Transformer-based baselines used official implementations with adapted hyperparameters. All methods were evaluated using the same data split and evaluation protocol.
> ### B.5 Additional Baselines and Comparisons
> **Question:** *Can the authors add or commit to adding direct comparisons with topology-aware losses (clDice, TopoLoss) and recent architectures?*
> **Response:** We have added additional comparisons with topology-aware losses (clDice and persistent homology loss) as well as recent architectures (SwinUNETR-V2 and STU-Net), and the results have been added to **Table 2**.
> ### B.6 Statistical Robustness and Lesion-Size Sensitivity
> **Question:** *Do the reported results hold across multiple random seeds, and is lesion-size stratification available for ET segmentation?*
> **Response:** This question is addressed in **A.1** and **A.2**.
> ### B.7 Code Availability
> **Question:** *Please share code.*
> **Response:** This question is addressed in **A.4**.

---

### Author Rebuttal · Authors · 2026-01-24

**Rebuttal:**

We sincerely thank the reviewers for their insightful comments and constructive feedback, which have significantly helped improve the clarity, rigor, and presentation of our work.

We have submitted a revised version of the manuscript that addresses all raised points. The main changes include:

- **Improved architectural clarity** with a detailed specification table (**Section 4**)
- **Expanded discussion and clarification** of topology violations (**Section 4.2**)
- **Additional ablation analysis and explanations** (**Section 5.2**)

In particular, following the reviewers' requests for clarity and reproducibility, we now include explicit pseudo-code for computing topology violations.

**Supporting Material:**

/attachment/a4703ae5f7cc888033091fc95a6e3deff288338b.zip

---

### Comment · Area_Chair_L3fN · 2026-01-26
**Post-rebuttal discussion and final ratings**

Dear reviewers,

Thank you for providing your comments on the paper. The authors have replied, and possibly modified their paper as a result.

You now have until the 1st of February to discuss with authors via the forum, and provide your final rating. This can change or stay the same depending on the rebuttal and discussion.

Your review, discussion, and final rating will be taken into account for the meta-review.

Once again thank you very much for your help.

---

### Comment · Area_Chair_L3fN · 2026-02-02
**Please provide final rating - reminder**

Dear reviewers, the rebuttal and discussion periods are over, could you please submit your final ratings today if not done already?

To do so please use "edit" on your official review.

Thank you again for your precious help in evaluating the papers.

---

### Meta-Review · Area_Chair_L3fN · 2026-02-05

**Recommendation:** Accept (Poster)
**Confidence:** 4

**Metareview:**

Authors have improved the paper and supplied a comprehensive rebuttal. No major flaws remain in the current state and the reviewers are satisfied.

---

### Decision · Program_Chairs · 2026-02-13

Accept (Poster)